# On-demand male contraception via acute inhibition of soluble adenylyl cyclase

Melanie Balbach[1], Thomas Rossetti[1], Jacob Ferreira[1], Lubna Ghanem[1], Carla Ritagliati[1], Robert W. Myers[2], David J. Huggins[2,3], Clemens Steegborn [4], Ileana C. Miranda [5], Peter T. Meinke[1,2], Jochen Buck [1] ✉ & Lonny R. Levin [1]

Nearly half of all pregnancies are unintended; thus, existing family planning options are inadequate. For men, the only choices are condoms and vasectomy, and most current efforts to develop new contraceptives for men impact sperm development, meaning that contraception requires months of continuous pretreatment. Here, we provide proof-of-concept for an innovative strategy for on-demand contraception, where a man would take a birth control pill shortly before sex, only as needed. Soluble adenylyl cyclase (sAC) is essential for sperm motility and maturation. We show a single dose of a safe, acutely-acting sAC inhibitor with long residence time renders male mice temporarily infertile. Mice exhibit normal mating behavior, and full fertility returns the next day. These studies define sAC inhibitors as leads for on-demand contraceptives for men, and they provide in vivo proof-of-concept for previously untested paradigms in contraception; on-demand contraception after just a single dose and pharmacological contraception for men.

The global unintended pregnancy rate is ~50%[1], and it is even higher among adolescents in the United States[2]. At present, preventing unintended pregnancies is largely the province of women. Of all modern contraceptive methods available, all but two are for women. The contraceptive choices for men are condoms or vasectomy, both of which have limitations which make them unsuitable for many men. Developing new contraceptives for men is challenging since after puberty, a man produces about 1000 sperm per second, and a male contraceptive strategy needs to be sufficiently effective to prevent millions of sperm from fertilizing the oocyte.

Hormonal approaches can successfully block sperm production by interfering with spermatogenesis. Male contraceptive efforts based on various hormonal regimens achieved overall success rates of ~94% in clinical studies, but these efforts were abandoned due to unwanted side effects[3–8]. A hormonal strategy involving a topically applied nestorone-testosterone gel in an ongoing world-wide clinical study remains the most advanced effort to develop a male contraceptive[9].

Other strategies focus on non-hormonal targets identified as essential for spermatogenesis via gene distruption technologies. Among the most advanced of these efforts, a small molecule inhibitor against bromodomain testes-specific protein reversibly suppressed spermatogenetis in mice[10], but it remains unclear whether the testis-specific isoform can be selectively inhibited[11]. Similarly, a retinoic acid receptor antagonist rendered mice reversibly sterile by blocking sperm production[12], but because retinoic acid receptors have other functions[13], there are concerns that chronic use may lead to undesirable side effects.

Two other efforts to develop a male contraceptive block sperm functions. The herb Triptonide, originally identified from Chinese folklore, caused sperm deformation in mice and nonhuman primates, resulting in infertile animals[14], and inhibitors of the epididymal peptidase inhibitor (EPPIN) block sperm functions in non-human primates[15,16]. All of these hormonal and non-hormonal strategies require months of continuous treatment prior to being effective[17], and they demand a similar time upon cessation of treatment to be fully

[1]Department of Pharmacology, Weill Cornell Medicine, New York, NY, USA. [2]Tri-Institutional Therapeutics Discovery Institute, New York, NY, USA. [3]Department of Physiology and Biophysics, Weill Cornell Medicine, New York, NY, USA. [4]Department of Biochemistry, University of Bayreuth, Bayreuth, Germany. [5]Laboratory of Comparative Pathology, Weill Cornell Medicine, Memorial Sloan Kettering Cancer Center, and The Rockefeller University, New York, NY, USA. ✉e-mail: jobuck@med.cornell.edu

reversible. Thus, the contraceptive development landscape lacks an on-demand approach where a contraceptive can be taken shortly before sexual intercourse and its effect is transient.

Morphologically mature mammalian sperm are stored in a dormant state within the cauda epididymis where the bicarbonate concentration is actively maintained at ≤5 mM[18]. Upon ejaculation, mixing with semen exposes the sperm to bicarbonate levels (~25 mM) which stimulate activity of bicarbonate-regulated soluble adenylyl cyclase (sAC; *ADCY10*). This sAC-dependent increase of cAMP is the initial signaling event activating sperm motility and capacitation[19–22], which are prerequisites for sperm to attain fertilizing capacity (reviewed in refs. [23–25]). sAC knockout (sAC KO) mice exhibit male-specific sterility[21,22,26], and two otherwise healthy men homozygous for mutations in the sAC gene (*ADCY10^{−/−}*) are sterile[19].

While sAC is found in a number of mammalian tissues, there is evidence that sAC isoforms present in sperm may be distinct from the more widely expressed isoforms[27]. Besides male-specific sterility, both sAC KO mice and *ADCY10^{−/−}* men exhibit few other phenotypes. In sAC KO mice, their reported elevated intraocular pressure[28] could lead to glaucoma, but only if elevated pressure is chronic. Similarly, while *ADCY10^{−/−}* men display an increased propensity to form kidney stones[19], presumably those can only form during prolonged periods of sAC absence. Historically, for contraceptive development, only targets in sperm were pursued that were exclusively expressed in testis[29,30]. However, these sAC null phenotypes suggest that contraception for men based on sAC inhibition could be safe and effective if sAC is only transiently blocked[27].

In mammals, two families of adenylyl cyclases produce cAMP: bicarbonate-regulated sAC and G-protein-regulated transmembrane adenylyl cyclases (tmACs)[31]. Among mammalian adenylyl cyclases, sAC is the evolutionarily ancient form, more closely related to adenylyl cyclases from cyanobacteria than it is to mammalian tmACs[32]. We identified numerous small molecule inhibitors which can selectively target sAC versus tmACs[22,33,34], and using these inhibitors in vitro, we validated that sAC inhibitors can be delivered intravaginally as a novel strategy for non-hormonal contraception in women[20]. Recently, starting from a non-toxic allosteric inhibitor that binds to sAC with micromolar potency identified in a high throughput screen[35], we used structure-assisted drug design to develop prototypes of increasingly potent sAC inhibitors with drug-like properties suitable for use in vivo to interrogate functions of sAC in animal models, diminished cross-reactivity with other mammalian nucleotidyl cyclases, and no significant cytotoxicity[36–38].

Here, we now show that administration of a single dose of an acutely-acting sAC inhibitor into male mice rapidly and temporarily inactivates sperm movement and renders the mice temporarily infertile. These data validate an effective on-demand contraception strategy unlike any other currently used form of birth control which avoids potential consequences of chronic dosing.

## Results

### A safe and effective sAC inhibitor with long residence time

We previously demonstrated that small molecule inhibitors which selectively target sAC[33,39] block numerous processes in mouse and human sperm essential for fertilization in vitro[20,22,35]. To explore the contraceptive potential via systemic delivery in males, a safe sAC inhibitor must retain its efficacy post-ejaculation, after semen and sperm are deposited into the inhibitor-free environment of the female reproductive tract. Inhibition with the previously used tool compound TDI-10229 did not survive dilution into inhibitor-free media (i.e., the sAC-inhibitor complex dissociated rapidly, releasing active enzyme)[20], consistent with TDI-10229 having a short residence time on sAC protein[38]. Our structure-assisted drug design efforts identified a more potent sAC inhibitor with longer residence time and drug-like properties suitable for use in vivo to interrogate functions of sAC in animal

models[37]. TDI-11861 (Fig. 1a) shows the same general binding mode as its parent compounds LRE1 and TDI-10229 (Fig. 1a) ($IC_{50}$ 159 nM). However, in addition to exploiting the bicarbonate binding site engaged by LRE1 and TDI-10229, TDI-11861 also occupies a channel leading to the active site (Fig. 1b)[35,37]. Thus, TDI-11861 binds sAC protein tighter and inhibits the in vitro adenylyl cyclase activity of purified sAC protein with an improved $IC_{50}$ of 3 nM (Fig. 1c) and sAC-specific cAMP accumulation in a cellular context with an $IC_{50}$ of 7 nM (Fig. 1d). TDI-11861 is highly specific for sAC; it had no activity against tmACs from each subfamily[37].

We used Surface Plasmon Resonance (SPR), where inhibitor solutions are flowed over a chip containing immobilized recombinant human sAC protein, to assess binding kinetics [i.e., rate constants for ligand association ($k_{(on)}$) and dissociation ($k_{(off)}$)]. The previously used tool compound TDI-10229 has a residence time ($1/k_{off}$) of 18 seconds (Fig. 1e), while TDI-11861 displays a significantly longer residence time of 61.5 minutes (Fig. 1f)[38]. Thus, in addition to being ~50 times more potent than TDI-10229 at inhibiting sAC protein in vitro and ~15 times more potent in cellular assays, TDI-11861 has the benefit of a ~200-fold longer residence time on sAC protein.

Like its predecessor TDI-10229[20,36], in addition to being specific for sAC versus related tmACs, sAC showed an overall benign safety profile, both in vitro and in vivo[37]. TDI-11861 had no significant activity (1) versus a 46-member safety panel comprised of ion channels, GPCRs, kinases, and nuclear receptors; (2) against over 310 kinases; (3) in hERG electrophysiology studies; (4) in glutathione trapping assays; (5) in AMES assays and DEREK analysis for mutagenicity risks; nor (6) cytotoxicity in cell viability assays at 20 μM, which is >2000 fold above its $IC_{50}$.

Our goal is acute sAC inhibition to provide on-demand contraception only when it is needed, and administration of a single oral dose of TDI-11861 did not cause any abnormal mouse behavior for at least 24 hours. Because a sAC inhibitor on-demand contraceptive would be subject to repeated dosing, we also explored the potential for mechanism-based toxicity due to long-term sAC inhibition. Unfortunately, limited solubility of TDI-11861 made it unsuitable for chronic delivery to mice. Instead, we leveraged a different long off-rate compound (TDI-11155) whose potency is similar to TDI-11861, and its solubility permitted use of osmotic minipumps for prolonged sAC inhibitor delivery. TDI-11155 (Fig. S2a) is a sAC-selective inhibitor with an $IC_{50}$ of 16 nM in cellular assays and whose $K_{on}$, residence time, and $K_i$ values are comparable to TDI-11861 (for TDI-11155, $K_{on} = 1.9 \times 10^5$/ms; residence time = 1474 sec; and $K_i = 3.8$ nM: For TDI-11861, $K_{on} = 2.1 \times 10^5$/ms; residence time = 2219 sec; and $K_i = 2.5$ nM)[38]. We delivered TDI-11155 for 7 days at levels 15-fold higher than its cellular $IC_{50}$ or for 42 days at levels threefold above its cellular $IC_{50}$ (Fig. S2b). In both cases, there were no observed behavioral effects during treatment, and no gross abnormalities were observed at study termination in any major organs, including testis and epididymis. At the histologic level, there was no histomorphologic evidence of sAC inhibitor-related toxicity in testis, epididymis, kidneys, eyes, liver, spleen, or pancreas (Supplemental Table S1). As mentioned above, sAC KO sterile men exhibit increased propensity to form kidney stones, yet we observed no signs of salt precipitation in mouse kidneys following chronic treatment. Thus, there appears to be no mechanism-based toxicity due to prolonged exposure to a sAC inhibitor.

As expected, and like TDI-10229[20] and other sAC inhibitors[22,35], treatment of sperm with TDI-11861 in vitro blocks capacitation, the maturation process sperm must undergo to gain fertilizing capacity. We mimicked capacitation in vitro by incubating sperm in buffer containing 25 mM bicarbonate and analyzed prototypical hallmarks of capacitation in the presence and absence of sAC inhibitors. In line with its higher potency on sAC protein (Fig. 1c) and in cellular assays (Fig. 1d), TDI-11861 was more potent than TDI-10229 at blocking the bicarbonate-induced cAMP rise in mouse and human sperm (Fig. 2a, c).

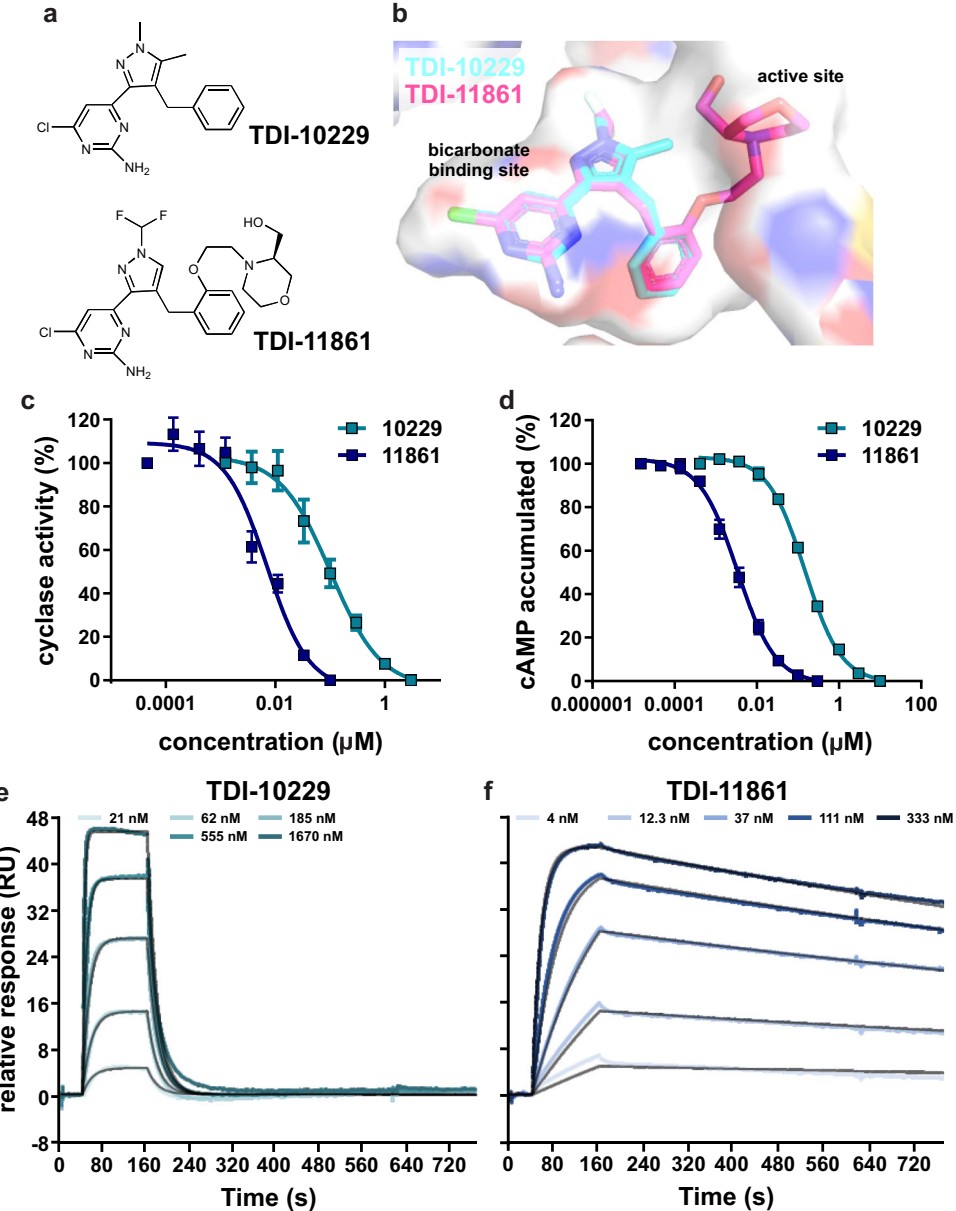

**Fig. 1 | A potent sAC inhibitor with longer residence time. a** Molecular structure of TDI-10229 and TDI-11861. **b** Overlay of the sAC complexes with TDI-10229 (cyan sticks) and TDI-11861 (magenta sticks). sAC, shown contoured as a molecular surface colored according to atom type, is from the TDI-11861 complex. **c** Concentration-response curves of TDI-10229 (teal) (IC$_{50}$ = 160 nM) and TDI-11861 (blue) (IC$_{50}$ = 3 nM) on in vitro adenylyl cyclase activity of purified recombinant human sAC protein in the presence of 1 mM ATP, 2 mM Ca$^{2+}$, 4 mM Mg$^{2+}$, and 40 mM HCO$_3^-$, normalized to the respective DMSO-treated control; mean ± SEM (*n* = 6 with individual replicates indicated as symbols). **d** Concentration-response curves of TDI-10229 (teal) (IC$_{50}$ = 102 nM) and TDI-11861 (blue) (IC$_{50}$ = 7 nM) on sAC-dependent cAMP accumulation in sAC-overexpressing rat

4-4 cells grown in media containing 10% FBS treated with 500 µM IBMX for 5 min, normalized to the respective DMSO-treated control; mean ± SEM (TDI-10229: *n* = 13, TDI-11871: *n* = 8 with individual replicates indicated as symbols). **e, f** Sensorgrams of **e** TDI-10229 (teal) or **f** TDI-11861 (blue) binding to immobilized human sAC protein measured using surface plasmon resonance. Representative traces of experiments repeated at least three times showing binding kinetics of different concentrations of inhibitor (colored lines) along with best fits using a 1:1 binding model (black lines). TDI-10229: $k_{on}$ = 2.3 × 10$^5$/ms, $K_D$ = 176 nM, $k_{off}$ = 55.8 × 10$^{-3}$/s; TDI-11861: $k_{on}$ = 2.1 × 10$^5$/ms, $K_D$ = 1.4 nM, $k_{off}$ = 0.3 × 10$^{-3}$/s. Source data are provided in the Source Data file, *n* = biological replicates.

Moreover, consistent with its longer residence time on sAC protein, TDI-11861 retained its ability to inhibit bicarbonate-induced cAMP synthesis in sperm following 100-fold dilution into inhibitor-free media (Fig. 2b, d). In contrast, TDI-10229 only inhibited the capacitation-induced cAMP rise in sperm when compound was present in the media.

Subsequent to the elevation of cAMP, two functional hallmarks of mammalian capacitation are increased flagellar beat frequency and the ability to undergo a physiologically induced acrosome reaction. As expected, TDI-11861 was also more potent than TDI-10229 at blocking

the bicarbonate-induced increase in flagellar beat frequency in both mouse (Fig. 2e, S3a, Supplementary Movie 1, Table S2) and human sperm (Fig. 2f, S3b, Supplementary Movie 2, Table S2) and the acrosome reaction induced by zona pellucidae in mouse sperm (Fig. 2g) or by progesterone in human sperm (Fig. 2h). As shown previously for TDI-10229[20], TDI-11861 was not toxic to sperm; addition of exogenous, cell-permeable cAMP/IBMX rescued the acrosome response blocked by sAC inhibition (Fig. 2g, h). Thus, TDI-11861 is more potent and displays a longer residence time than TDI-10229 at blocking capacitation-induced changes when added exogenously to sperm. Therefore,

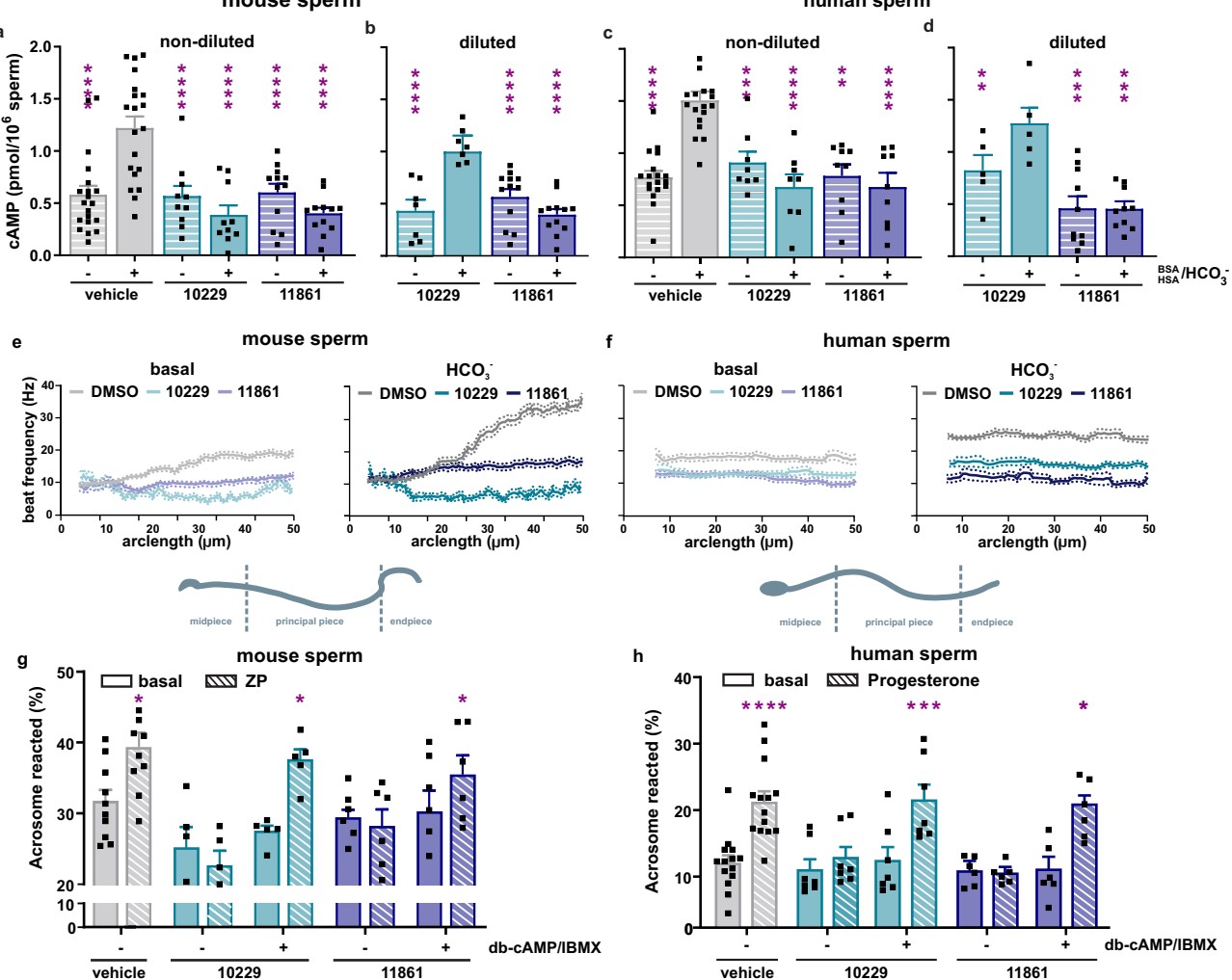

**Fig. 2 | TDI-11861 inhibits essential functions required for fertilization in mouse and human sperm. a**, **c** Intracellular cAMP levels in **a** mouse and **c** human sperm incubated in non-capacitating (striped bars) or capacitating media (solid bars) in the absence (grey) or presence of 5 µM TDI-10229 (teal) or 10 nM TDI-11861 (blue). Shown are cAMP levels measured after 12 (mouse) or 30 minutes (human); mean + SEM (mouse: vehicle $n = 20$, TDI-10229 $n = 10$, TDI-11861 $n = 12$; human: vehicle $n = 18$, TDI-10229 $n = 8$, TDI-11861 $n = 9$ with individual replicates indicated as symbols). **b**, **d** Intracellular cAMP levels in **b** mouse and **d** human sperm following dilution into inhibitor-free media. After preincubation (5 min) in 5 µM TDI-10229 (teal) or 10 nM TDI-11861 (blue), sperm were diluted (1:100) in inhibitor-free non-capacitating (striped bars) or capacitating media (solid bars). Shown are cAMP levels measured 12 (mouse) or 30 (human) minutes after dilution; mean + SEM (mouse: TDI-10229 $n = 7$, TDI-11861 $n = 12$; human: TDI-10229 $n = 5$, TDI-11861 $n = 10$ with individual replicates indicated as symbols). **e**, **f** Mean flagellar beat frequency along the length of the tail (arc length, µm) of **e** mouse and **f** human sperm in the absence (grey) or presence of 5 µM TDI-10229 (teal) or 10 nM TDI-11861 (blue) before and after stimulation with 25 mM NaHCO3. Solid lines indicate the time-averaged values, dotted lines the SEM, $n = 3$, ≥60 individual sperm from three different mice or three different human donors. **g**, **h** Acrosome reaction in **g** mouse sperm evoked by 50 heat-solubilized zona pellucidae (striped bars) and **h** human sperm evoked by 10 µM progesterone (striped bars) after incubation for 90 min (mouse) or 180 min (human) in capacitating media in the absence (grey) or presence of 5 µM TDI-10229 (teal) or 10 nM TDI-11861 (blue) in the absence or presence of 5 mM db-cAMP (dibutyrl-cAMP)/500 µM IBMX (isobutylmethylxanthine); mean + SEM (mouse: vehicle $n = 10$, TDI-10229 $n = 5$, TDI-11861 $n = 6$; human: vehicle $n = 14$, TDI-10229 $n = 7$, TDI-11861 $n = 6$ with individual replicates indicated as symbols). ZP = zona pellucidae. Differences between conditions were analyzed using one-way ANOVA compared to the DMSO-treated capacitated control *$P < 0.05$, **$P < 0.01$, ***$P < 0.001$, ****$P < 0.0001$. Source data are provided in the Source Data file, $n$ = biological replicates.

TDI-11861 represents a suitable tool compound for studying sAC biology in sperm in vivo; it is safe, effective in vitro, and has a longer residence time on sAC protein to counteract the dilution effects in the inhibitor-free female genital tract.

## A single oral dose of sAC inhibitor blocks essential sperm functions

As with TDI-10229[20], oral administration of TDI-11861 (50 mg/kg) via gavage resulted in blood levels at least 100-fold higher than the inhibitor's cellular IC$_{50}$ with fast onset post administration (Fig. S4). To test whether orally administering sAC inhibitors into male mice will impact sperm functions in vivo, we assessed essential functions

(i.e., capacitation-induced, sAC-dependent cAMP rise and motility) in sperm surgically extracted from the cauda epididymis one hour after orally delivering either TDI-10229 (50 mg/kg) or TDI-11861 (50 mg/kg). At one and three hours post oral administration, levels of TDI-11861 in various tissues, including male reproductive organs (i.e., both caput and cauda epididymis and seminal vesicles, but notably not testis), correlated well with its concentration in blood (Fig. S5). When we measured cAMP responses in sperm directly extracted from the cauda one hour after oral administration, sperm from mice receiving either TDI-10229 or TDI-11861 displayed no bicarbonate-induced increase in cAMP (Fig. 3a). However, when we diluted the isolated epididymal sperm 100-fold ex vivo, to imitate the dilution into the inhibitor-free

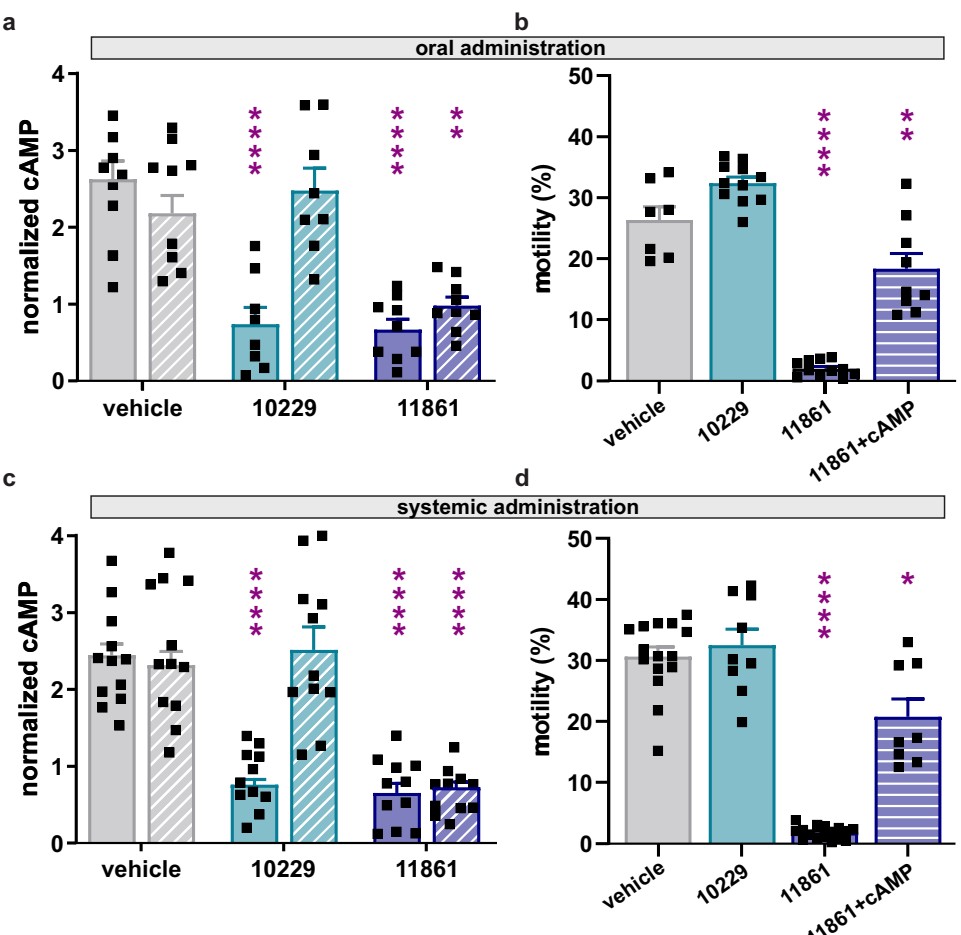

**Fig. 3 | A single dose of systemically delivered sAC inhibitors block essential mouse sperm functions ex vivo. a, c** Relative cAMP increase due to incubation in capacitating conditions of epididymal mouse sperm isolated at the indicated times following (a) oral administration (gavage) or (c) injection (i.p.) with vehicle (DMSO:PEG 400:PBS 1:4:5) (grey), 50 mg/kg TDI-10229 (teal) or 50 mg/kg TDI-11861 (blue). Isolated sperm were measured directly (solid bars) or after 100-fold dilution (striped bars) into inhibitor-free non-capacitating or capacitating media, and cAMP was measured after incubation for 12 minutes. Values shown are cAMP levels in capacitating sperm relative to cAMP levels in non-capacitated sperm from the same mouse; mean + SEM (oral: vehicle $n = 9$, TDI-10229 $n = 8$, TDI-11861 $n = 9$; systemic: vehicle $n = 12$, TDI-10229 $n = 11$, TDI-11861 $n = 11$ with individual replicates indicated as symbols). **b, d** Percentage of motile epididymal mouse sperm isolated 1 hour post (b) oral administration (gavage) or (d) injection (i.p.) with vehicle (grey), 50 mg/kg TDI-10229 (teal) or 50 mg/kg TDI-11861 (blue). Isolated sperm were diluted 1:100 in inhibitor-free non-capacitating media, and percent motility was assessed by computer-assisted sperm analysis (CASA). For sperm isolated from TDI-11861-injected males, motility was also assessed in the presence of 5 mM db-cAMP/500 µM IBMX (striped bars); mean + SEM (oral: vehicle $n = 7$, TDI-10229 $n = 11$, TDI-11861 $n = 11$, TDI-11861+cAMP = 9; systemic: vehicle $n = 15$, TDI-10229 $n = 9$, TDI-11861 $n = 16$, TDI-11861+cAMP = 8 with individual replicates indicated as symbols). Differences between conditions were analyzed using one-way ANOVA compared to sperm isolated from vehicle-injected mice, *$P < 0.05$, **$P < 0.01$, ****$P < 0.0001$. Source data are provided in the Source Data file, $n$ = biological replicates.

female reproductive tract, sperm from mice treated with the short residence time inhibitor TDI-10229 recovered bicarbonate-induced cAMP responsiveness, while sperm from mice treated with the longer residence time inhibitor TDI-11861 remained non-responsive to bicarbonate.

We next assessed the effects of orally delivered sAC inhibitors on motility of cauda epididymal sperm. Sperm from sAC KO mice and infertile men with sAC mutations show only small vibratory movements[19–22,26]. Similarly, sperm from the epididymis of orally administered, TDI-11861-treated mice were essentially immotile (Fig. 3b, Fig. S6, Supplementary Movie 3), displaying only vibratory movements reminiscent of sperm from sAC KO mice[20–22] and men[19]. Because sperm isolated from the epididymis are diluted ex vivo to assess motility, sperm from mice treated with the shorter residence time inhibitor, TDI-10229, were motile and indistinguishable from sperm from vehicle-treated mice. As expected, addition of exogenous membrane-permeable cAMP rescued motility in TDI-11861-treated sperm, confirming that TDI-11861 is not cytotoxic to sperm and

functions via inhibiting sAC. These data validate the feasibility of on-demand male contraception; a single oral dose of a long-residence time sAC inhibitor to a male mouse acutely blocks the cAMP-mediated sperm functions required for fertility, and its effects persist through ex vivo dilution.

To assess the in vivo contraceptive effects of sAC inhibitors in larger numbers of animals, we changed to a more facile route of administration (i.e., intraperitoneal (i.p.) injection). Similar to orally delivered TDI-11861, levels of i.p.-injected TDI-11861 in tissues, including cauda epididymis but excluding testis, correlated well with levels in blood (Fig. S5). As expected, the in vivo effects of injecting sAC inhibitors on sperm function mirrored their effects when compounds were administered via oral gavage. Both TDI-10029 (50 mg/kg) and TDI-11861 (50 mg/kg) prevented bicarbonate-induced increase in cAMP in sperm directly isolated from cauda epididymis (Fig. 3c), but only TDI-11861 inhibition of the bicarbonate-induced elevation of cAMP (Fig. 3c) and motility (Fig. 3d, Fig. S7, Supplementary Movie 4, Supplementary Movie 5, Supplementary Movie 6) persisted following ex vivo dilution.

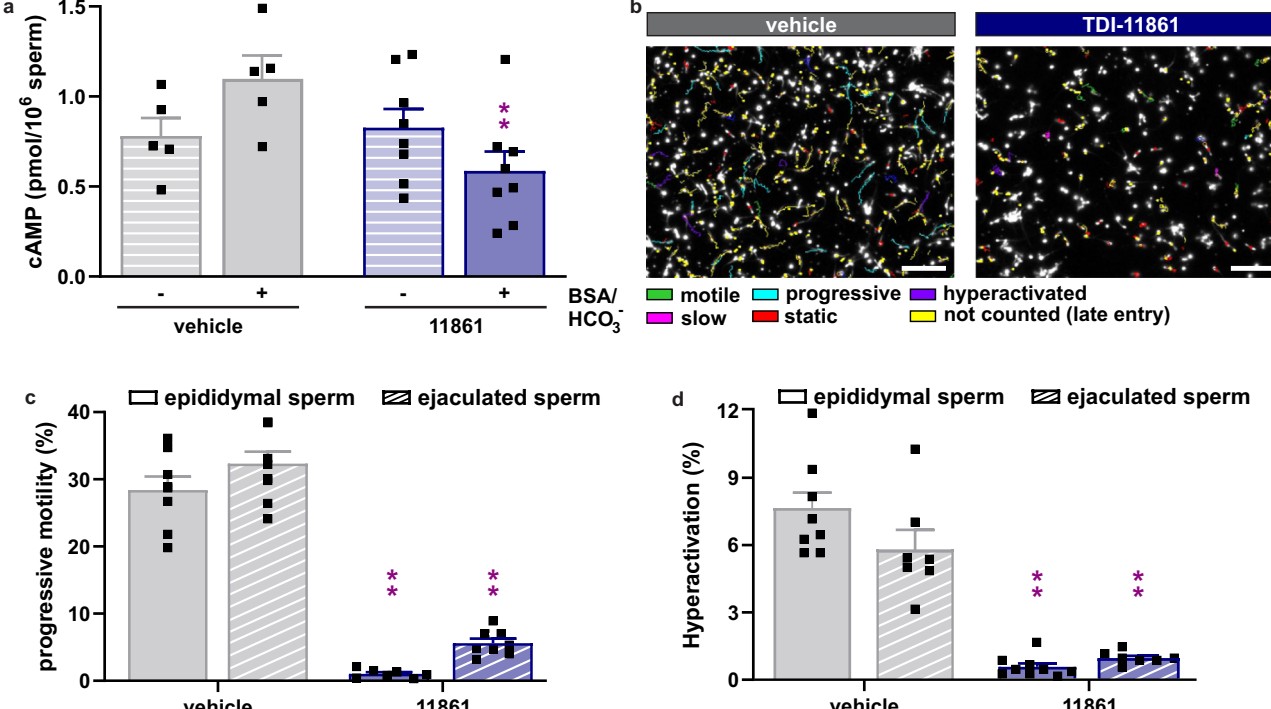

**Fig. 4 | Systemically delivered TDI-11861 blocks motility and capacitation of ejaculated mouse sperm recovered post-coitally from the uterus. a** Intracellular cAMP levels in ejaculated uterine mouse sperm isolated 2 hours following injection (i.p.) with vehicle (grey) or 50 mg/kg TDI-11861 (blue) incubated in non-capacitating (striped bars) or capacitating media (solid bars); mean + SEM (vehicle $n = 5$, TDI-11861 $n = 8$ with individual replicates indicated as symbols). **b** Representative motility tracks of ejaculated uterine mouse sperm isolated 2 hours following injection (i.p.) with vehicle or 50 mg/kg TDI-11861. Scale bar = 300 μM. Track´s color code: motile (green), progressive (turquoise), hyperactivated (purple), slow (pink), static (red), not counted due to late entry (yellow). **c, d** Percentage of (c) progressively motile and (d) hyperactivated epididymal (solid bars) and ejaculated uterine (striped bars) mouse sperm isolated two-hour post injection (i.p.) with vehicle (grey) or 50 mg/kg TDI-11861 (blue); mean + SEM (progressive: vehicle epididymal $n = 8$, TDI-11861 epididymal $n = 7$, vehicle ejaculated $n = 9$, TDI-11861 ejaculated $n = 8$, hyperactivated: vehicle epididymal $n = 8$, TDI-11861 epididymal $n = 9$, vehicle ejaculated $n = 7$, TDI-11861 ejaculated $n = 7$ with individual replicates indicated as symbols). Differences between vehicle-injected and TDI-11861-injected sperm were analyzed using two-tailed, unpaired $t$ test, **$P < 0.01$. Source data are provided in the Source Data file, $n$ = biological replicates.

We also examined the in vivo effects of TDI-11861 by studying ejaculated sperm from injected males retrieved one-hour post-coitus from the uteri of recipient females[40]. TDI-11861-treated males mated with, and deposited sperm into, receptive females with the same frequency as vehicle-injected males. Thus, sAC inhibitor treatment does not affect mating behavior or ejaculation. However, as seen in their epididymal sperm, ejaculated sperm from TDI-11861-treated mice were immotile and their capacitation-induced effects were blocked (Fig. 4). As expected, in ejaculated sperm from vehicle-injected mice, exogenous bicarbonate supported both cAMP production (Fig. 4a) as well as progressive motility (Fig. 4c) and the prototypical change to hyperactivated motility (Fig. 4d), which is the vigorous, asynchronous beating pattern necessary for sperm to reach and penetrate the oocyte's vestments. In contrast, TDI-11861-treated sperm retrieved from the uterus showed no bicarbonate-induced increase in cAMP (Fig. 4a) and few sperm were motile (Fig. 4b, c, Supplementary Movie 7) or hyperactivated (Fig. 4d). Thus, a single dose of a long-residence time sAC inhibitor given to male mice blocks motility and capacitation in sperm ejaculated into the female genital tract without affecting mating behavior.

We used motility of epididymal sperm diluted ex vivo to explore the time course of TDI-11861 inhibition and assess the relationship between sAC inhibitor pharmacokinetics (PK) and sperm function essential for fertility. TDI-11861 inhibited motility as early as 15 minutes post i.p.-injection when blood levels of TDI-11861 were 20 μM (Fig. 5). TDI-11861 blood levels decreased following first-order kinetics, and by 4 hours post-injection, the level of TDI-11861 was 1 μM. A subset of

sperm regained ex vivo motility (i.e., subsequent to dilution after surgical extraction from epididymis) three hours post-injection with increasing number of sperm regaining motility as TDI-11861 blood levels decreased. By 24 hours post-injection, most sperm from TDI-11861-injected mice had recovered motility (Supplementary Movie 6).

## TDI-11861 blocks fertility in vivo

Because a single injection of TDI-11861 (50 mg/kg) stably inhibits motility for up to 2.5 hours, we explored its contraceptive efficacy over this time frame in timed mating studies. When vehicle-injected mice were paired with receptive females (i.e., females visually identified to be in estrus) from 30 minutes post injection through 2.5 hours post injection (i.e., 2-hour pairing), they impregnated females 30% of the time. In contrast, over this same time frame, TDI-11861-injected males were completely infertile; zero females became pregnant in 52 pairings (Table 1). As mentioned above, injection with TDI-11861 did not adversely affect mouse behavior; (1) movements and mating behaviors were indistinguishable between TDI-11861 and vehicle-injected males; (2) equivalent percentages (~24%) of females exhibited vaginal plugs in both mating groups; and (3) recovery of ejaculated sperm from the uteri of mated females (Fig. 4) confirmed that the TDI-11861-injected males copulated and ejaculated normally.

Predictably, as we extended the pairing time beyond the point where sperm from injected mice started to recover ex vivo motility (Fig. 5), contraceptive efficacy decreased. There was a single pregnancy in 45 three-hour pairings (i.e., from 0.5 hours to 3.5 hours post injection) and 5 pregnancies out of 55 eight-hour pairings (i.e., from

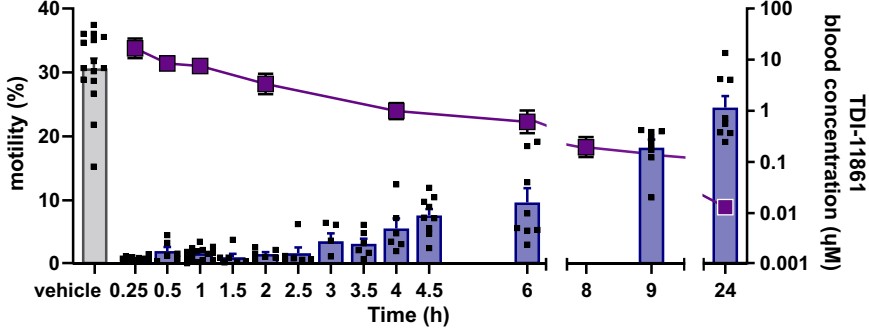

**Fig. 5 | TDI-11861 inhibition of sperm motility correlates with pharmacokinetics.** Percentage of motile epididymal mouse sperm (left axis and bars; mean + SEM; vehicle (grey) $n = 15$, TDI-11861 (blue) 0.25h $n = 10$, 0.5h 1.5h 2h 2.5h 3.5h 4h $n = 6$, 1h $n = 16$, 3h $n = 5$, 4.5 h $n = 9$, 6h 9h 24h $n = 8$ with individual replicates indicated as symbols) and TDI-11861 blood concentration (right axis and blue squares; mean ± SEM; $n = 5$) quantified at the indicated time points post injection (i.p.) with vehicle or 50 mg/kg TDI-11861. Source data are provided in the Source Data file, $n$ = biological replicates.

**Table 1 | sAC inhibitors inhibit fertility of male mice in timed matings**

| Mating period (hours post injection) | Mating time (hours) | Vehicle | | TDI-11861 | | |
|---|---|---|---|---|---|---|
| | | Pregnancy rate | Pregnancy pairings | Pregnancy rate | Pregnancy pairings | Contraceptive efficacy |
| 0.5–2.5 | 2 | 30% | (15/50) | 0% | (0/52) | 100% |
| 0.5–3.5 | 3 | 24% | (11/45) | 2.2% | (1/45) | 91% |
| 1–9 | 8 | 41% | (39/95) | 9.1% | (5/55) | 78% |
| 24–120 | 96 | 93% | (14/15) | 87% | (13/15) | |

Pregnancies in %, number of pregnancies per total amount of pairings, and contraceptive efficacy in % compared to vehicle-injected control matings using sAC inhibitor-injected males. Contraceptive efficacy is defined as the pregnancy rate in sAC inhibitor-injected mice divided by the pregnancy rate in vehicle-injected mice subtracted from 100%. Vehicle- or 50 mg/kg TDI-11861-injected males were mated with sexually receptive non-injected females for the indicated time periods. 24 hours after vehicle or TDI-11861 injection, 15 randomly chosen males were mated with females for 4 days.

1 hour to 9 hours post injection) (Table 1). Extending the pairing times afforded us the opportunity to examine the health of the offspring resulting from breakthrough pregnancies (i.e., from sperm previously inhibited by TDI-11861). Both male and female F1 progeny from breakthrough pregnancies matured normally into fertile adults. Furthermore, we found no evidence that reduced pregnancies from sAC inhibitor injected males were due to abortifacient activity. When pregnancies were assessed by uterine inspection 7 days post-mating, there were no signs of aborted fetuses in females mated with either sAC inhibitor- or vehicle-treated males. Finally, effects on fertility were fully reversible. 24 hours after injection, randomly chosen TDI-11861-injected males were mated with females. To maximize mating efficiency males were paired with females for 4 days, and 93% of the pairings yielded litters (Table 1).

## Discussion

Here, we employed an acutely acting sAC-specific inhibitor with a long residence time on sAC protein to provide in vivo proof-of-concept that sAC inhibitors can be developed into safe, non-hormonal, on-demand, male contraceptives. These studies demonstrate feasibility of two ground-breaking paradigms for human contraception; non-hormonal male contraception and on-demand pharmacological contraception.

A major distinction between mouse and human reproduction is the female anatomy. In mice, there is no physical barrier between the vagina and uterus, and semen deposited in the vagina advances unimpeded into the uterus[41]. In humans, ejaculated sperm must be motile to cross the cervix[42] to escape the normally inhospitable environment of the vagina and enter the permissive environment of the uterus. Once sperm cross the cervix, they can persist for days, allowing human conception to occur days following copulation[43]. Because sAC inhibitors, delivered to male mice via either oral gavage or i.p. injection, block progressive and hyperactivated motility, sperm from a man who has taken a long residence time sAC inhibitor contraceptive should not be able to cross the cervix and will remain trapped in the vagina. As the vagina re-acidifies soon after copulation[44], sAC-inhibited sperm would not persist long after sex.

The on-demand strategy we validate here, where a man will be temporarily infertile shortly after taking a single dose of a contraceptive agent, is qualitatively distinct from all other existing pharmacologic methods as well as known efforts to develop a male contraceptive (reviewed in[30,45–48]). For women, all pharmacologic strategies depend upon hormonal agents, which require continuous treatment to achieve contraceptive efficacy. Similarly, hormonal strategies for men currently in clinical trials work by blocking testicular sperm production. These methods[49], as well as other experimentally validated hormonal and non-hormonal strategies which affect spermatogenesis[10,12,50,51], require months of continuous usage before sperm numbers fall to subfertile levels. These methods also need months after cessation of therapy to recover normal sperm counts. Two post-spermatogenesis, non-hormonal strategies for male contraception have been validated in animal models, and while these do not depend upon disrupting sperm production, they also require chronic treatment[14–16]. Unlike these methods, on-demand contraception with a sAC inhibitor rendered male mice infertile within 30 minutes of a single dose of inhibitor, and fertility was fully restored the next day.

In addition to being more convenient, therapeutics acting acutely are less likely to elicit unwanted side effects relative to chronic treatments. Specifically for sAC, besides male-specific infertility, the phenotypes observed in mice or men in the absence of sAC require long periods of time to manifest[27]. Elevated intraocular pressure requires years to cause glaucoma[28], and kidney stones will only form after prolonged absence of sAC. Plus, there is precedent for an on-demand therapeutic targeting a broadly expressed target that is safely administered and widely adopted. Like sAC, the target of erectile dysfunction therapeutics, the cGMP-specific phosphodiesterase 5 (PDE5), is widely expressed[52], yet acute PDE5 inhibitors (i.e., sildenafil, vardenafil, tadalafil) are sufficiently safe for widespread use.

The mouse studies presented here leveraged a tool compound that blocked sperm functions in vivo whether delivered orally or via i.p. injection. These studies provide a framework for developing an on-demand male contraceptive and define sAC inhibitors as lead compounds for on-demand, non-hormonal, male contraceptives. A sAC inhibitor with suitable pharmacokinetics, oral bioavailability, long residence time, and safety profile can be formulated into an oral male birth control pill, which a man would take shortly before sex, providing protection from unwanted pregnancy for the subsequent hours. This innovative, on-demand, non-hormonal strategy represents a previously untested concept in contraception, which has the potential to provide equity between the sexes and, like the advent of oral birth control for women, revolutionize family planning.

## Methods

### Inclusion and ethics statement

This research posed no health, safety, security, or other risk to researchers. Animal experiments were approved by Weill Cornell Medicine's Institutional Animal Care and Use Committee (IACUC). Only males can be sperm donors. Samples of human semen were obtained from healthy volunteers reflecting the ethnic makeup of the Weill Cornell community with their prior written consent following a protocol approved by Weill Cornell Medicine's Institutional Review Board (IRB 21-03023495).

### Reagents, cell lines, and mice

3-Isobutyl-1-methylxanthine (IBMX), BSA, dibutyryl-cAMP (db-cAMP), hyaluronidase, lectin from *Pisum sativum* FITC-conjugated (PSA-FITC) and lectin from *Arachis hypogaea* FITC-conjugated (PNA-FITC) were purchased from Sigma-Aldrich, ionomycin from Tocris, β-mercaptoethanol from Gibco, and hormones from ProSpec. PBS buffer was purchased from Corning, DMEM, and 0.5 M EDTA, pH 8.0 from Thermo Fisher Scientific, FBS from Avantor Seradigm, and polyethylene glycol 400 (PEG 400) from Merck Millipore.

sAC-overexpressing rat 4-4 cells were generated and functionally authenticated in our laboratory as previously described[33] and grown in DMEM + 10% FBS. Cells were maintained at 37 °C in 5% $CO_2$ and were periodically checked for mycoplasma contamination.

8–10 week-old C57BL/6J male and female mice were purchased from Jackson Laboratories and allowed to acclimatize before use. All animals were maintained at a 12 hr light/dark cycle, temperatures of 64–79 °F (-18–26 °C) with 40–60% humidity in accordance with the NIH Guide for the Care and Use of Laboratory Animals. Animal experiments were approved by Weill Cornell Medicine's Institutional Animal Care and Use Committee (IACUC).

### Sperm isolation

Mouse sperm were isolated by incision of the cauda epididymis followed by 'swim-out' in 500 µl Toyoda Yokoyama Hoshi (TYH) medium (in mM: 135 NaCl, 4.7 KCl, 1.7 $CaCl_2$, 1.2 $KH_2PO_4$, 1.2 $MgSO_4$, 5.6 glucose, 0.56 pyruvate, 10 HEPES, pH 7.4 adjusted at 37 °C with NaOH), prewarmed to 37 °C. After 15 min swim-out at 37 °C, sperm from two caudae were combined and counted using a hematocytometer. For capacitation, sperm were incubated for 90 min in TYH containing 3 mg/ml BSA and 25 mM $NaHCO_3$ in a 37 °C incubator. To control for the consequences of dilution during isolation of epididymal mouse sperm for ex vivo assays, we assessed the ability of bicarbonate to induce a prototypical pattern of tyrosine phosphorylation (pY) which is a widely used molecular hallmark of capacitation[53]. Bicarbonate-induced pY is known to be sAC dependent in vitro[20,22,35]. For experiments studying sperm from injected mice, 'swim out' was performed in 200 µl TYH, which corresponded to a 1:10 dilution from epididymis (20 µg cauda in 200 µl buffer). Capacitation-induced changes were assessed by adding 50 µl of 'swim out' sperm to increasing volumes of capacitating TYH buffer. Sperm from vehicle-injected mice showed the

capacitation-induced increase in pY regardless of dilution in capacitation media (Fig. S1a, b). The pY pattern was blocked in sperm from both TDI-10229 and TDI-11861 injected mice when the 'swim out' sperm were minimally diluted by mixing with equal volume capacitation media (Fig. S1c–f). When diluted 25 fold, the pY pattern was restored in sperm isolated from mice injected with the short residence time inhibitor TDI-10229 (Fig. S1c, d). In contrast, the pY pattern remained blocked in 'swim out' using sperm from TDI-11861 injected mice even when they were diluted 100 fold (Fig. S1e, f). Because TDI-10229 inhibition survived the minimal dilution, but not the more substantial (i.e., 25-fold) dilution, we compared ex vivo bicarbonate-induced cAMP changes under these different conditions.

For analysis of ejaculated uterine mouse sperm, the uteri of females with mating plugs were removed and placed into 1 ml TYH buffer. The uteri were ruptured, and sperm were collected after 15 min swim-out at 37 °C.

Samples of human semen were obtained from healthy volunteers with their prior written consent following a protocol approved by Weill Cornell Medicine's Institutional Review Board (IRB 21-03023495). Only samples that met the WHO 2010 criteria for normal semen parameters (ejaculated volume ≥1.5 mL, sperm concentration ≥15 million/mL, motility ≥40%, progressive motility ≥32%, normal morphology ≥4%) were included in this study. Semen was incubated for 30 min in a 37 °C incubator to liquefy. Human sperm were purified by "swim-up" procedure in human tubular fluid (HTF) (in mM: 97.8 NaCl, 4.69 KCl, 0.2 $MgSO_4$, 0.37 $KH_2PO_4$, 2.04 $CaCl_2$, 0.33 Na-pyruvate, 2.78 glucose, 21 HEPES, pH 7.4 adjusted at 37 °C with NaOH). 0.5 to 1 ml of liquefied semen was layered in a 50 ml tube below 4 ml HTF. The tubes were incubated at a tilted angle of 45° at 37 °C for 60 min. Motile sperm were allowed to swim up into the HTF layer; immotile sperm and other cells or tissue debris remain in the ejaculate fraction. Up to 3 ml of the HTF layer was transferred to a fresh tube and washed twice in HTF by centrifugation ($700 \times g$, 20 min). The supernatant was removed after the last centrifugation step and the sperm pellet was resuspended in 1 ml HTF. The purity and vitality of each sample was assessed via light microscopy. Sperm cell numbers were determined using a hemocytometer and adjusted to a concentration of $1 \times 10^7$ cells/ml. For capacitation, sperm were incubated in HTF with 72.8 mM NaCl containing 25 mM $NaHCO_3$ and 3 mg/ml human serum albumin (HSA) (Irvine Scientific, Santa Ana, CA, USA) or 3 mg/ml BSA for up to 3 h.

### In vitro adenylyl cyclase activity assay

All in vitro adenylyl cyclase activity assays were performed via the "two-column" method measuring the conversion of [α-$^{32}$P] ATP into [$^{32}$P] cAMP[54], as previously described[55]. Briefly, human $sAC_t$ protein was incubated in buffer containing 50 mM Tris-HCl, pH 7.5, 4 mM $MgCl_2$, 2 mM $CaCl_2$, 1 mM ATP, 3 mM DTT, 40 mM $NaHCO_3$ in the presence of the indicated concentrations of different sAC inhibitors or vehicle (1% DMSO).

### Cellular cAMP accumulation

To assess potency in a cellular system, we utilized 4-4 cells, which stably overexpress sAC[20,33,56]. Cellular levels of cAMP reflect a balance between its synthesis by adenylyl cyclases and its degradation by phosphodiesterases (PDEs). Hence, in the presence of the non-selective PDE inhibitor IBMX, cells accumulate cAMP solely dependent upon the activity of endogenous adenylyl cyclases, which in 4-4 cells, is exclusively due to sAC[33,56]. On the day prior to the assay, $5 \times 10^6$ cells/ml were seeded in 24-well plates in DMEM with 10% FBS. To measure sAC-dependent cAMP accumulation, cells were pretreated for 10 min with the respective inhibitor at the indicated concentrations or DMSO as control in 300 µl fresh media. Cyclic AMP accumulation was initiated by the addition of 500 µM IBMX, and after 5 min, the media was removed, and the cells were lysed with 250 µl 0.1 M HCl by shaking at 700 rpm for 10 min. Cell lysates were centrifuged at $2000 \times g$ for

3 min and the cAMP in the supernatant was quantified using the Direct cAMP ELISA kit (Enzo) according to the manufacturer's instructions.

cAMP generation was measured in mouse and human sperm. For mouse sperm, aliquots of $2 \times 10^6$ cells were incubated for 12 min in the presence or absence of sAC inhibitor in non-capacitating or capacitating TYH buffer. For human sperm, aliquots of $2 \times 10^6$ cells were incubated for 30 min in the presence or absence of sAC inhibitor in non-capacitating or capacitating HTF buffer. In both cases, 0.1% DMSO was used as vehicle control.

For wash-out experiments assessing dilution of sAC inhibitors from sperm, sperm were pre-incubated for 5 minutes in non-capacitating media in the presence of sAC inhibitor at a concentration 5x above its $IC_{50}$. After 5 min, 150 µl of sperm/inhibitor mix was diluted into 1.35 ml non-capacitating or capacitating media with no inhibitor. After 12 min (mouse sperm) or 30 min (human sperm), sperm were sedimented by centrifugation at $2000 \times g$ for 3 min and lysed in 200 µl HCl for 10 min. Sperm lysates were centrifuged at $2000 \times g$ for 3 min and the cAMP in the supernatant was acetylated and quantified using the Direct cAMP ELISA kit (Enzo).

For ex vivo determination of sperm cAMP generation, 150 µl of solution containing 50 mg/kg sAC inhibitor were administered to male mice either orally via gavage or systemically via intraperitoneal (i.p.) injection; control males were subjected to 150 µl vehicle control (DMSO:PEG 400 1:4 (v/v) for TDI-10229, DMSO:PEG 400:PBS 1:4:5 (v/v) for TDI-11861). Sperm (50 µl) isolated at indicated time points (between 1 hour and 24 hours post-injection) were incubated in 50 µl (1:20 dilution) or 450 µl (1:200 dilution) non-capacitating or capacitating TYH media for 12 min. Ejaculated uterine mouse sperm were diluted 1:5 and incubated in non-capacitating or capacitating TYH for 30 min. Because samples of ejaculated mouse sperm collected from the uterus of impregnated females included endogenous uterine cells, basal cAMP levels reflect contributions from both. Intracellular cAMP levels were quantified using the Direct cAMP ELISA kit (Enzo) as above.

### Measuring sAC inhibitor binding kinetics using surface plasmon resonance

Association and dissociation rate constants of sAC inhibitors were obtained with a Biacore 8K instrument (Cytiva) using a parallel kinetics protocol. Series S Sensor NTA chips (Cytiva) were prepared by applying recombinant purified His-tagged $sAC_t$ protein (50 µg/ml) in PBS-P+ buffer (1 mM $KH_2PO_4$, 150 mM NaCl, 6 mM $Na_2HPO_4$, 0.05% (w/v) P20 Surfactant). The His-tagged sAC protein was captured via $Ni^{2+}$-His-tag chelation and covalently immobilized by amine coupling with a 1:1 mixture of 1-ethyl-3-(3-dimethylaminopropyl) carbodiimide and N-hydroxysuccinimide (active channels). After coupling, remaining reactive groups on the chip's surface were blocked with 1 M ethanolamine followed by 350 mM EDTA to wash away any free $Ni^{2+}$. Following preparation, TBS-P+ running buffer (50 mM Tris, 150 mM NaCl, and 0.05% P20 Surfactant) supplemented with 1% DMSO was flowed over the surface of the chip until a stable baseline was obtained. For each sAC inhibitor, which were dissolved in DMSO and diluted in running buffer, increasing concentrations were injected into parallel channels for 120 s at a flow rate of 50 µl/min followed by running buffer for 600 s to allow for dissociation. All experiments included reference channels; i.e., inhibitor run over parallel channels without immobilized protein. Binding kinetics were determined by subtracting responses in the reference channels from responses in the active channels. Curves were fitted, and $k_{on}$, $K_D$, and $k_{off}$ values were determined using the Biacore 8K Insight Evaluation Software Version 2.0 (Cytiva) and a 1:1 binding kinetics model.

### Isolation of mouse zonae pellucidae

*Zonae pellucidae* were isolated from female mice superovulated by intraperitoneal injection of 10 I.U. human chorionic gonadotropin 3 days before the experiment. 14 h before oocyte isolation, mice were

injected with 10 I.U. pregnant mare's serum gonadotropin. Mice were sacrificed by cervical dislocation and oviducts were dissected. Cumulus-enclosed oocytes were separated from the oviducts and placed into TYH buffer containing 300 µg/ml hyaluronidase. After 15 min, cumulus-free oocytes were transferred into fresh buffer and washed twice. Zonae pellucidae and oocytes were separated by shear forces generated by expulsion from 50 nM pasteur pipettes. Zona pellucidae were counted, transferred into fresh buffer, and heat-solubilized by incubation for 10 min at 65 °C.

### Acrosome reaction assay

For analysis of acrosomal exocytosis, 100 µl of $1 \times 10^7$ sperm/ml were capacitated for 90 min in TYH buffer supplemented with 3 mg/ml BSA and 25 mM $NaHCO_3$ (mouse sperm) or HTF buffer supplemented with 3 µl/ml HSA and 25 mM $NaHCO_3$ (human sperm). sAC inhibitors were added with capacitating buffer; 0.1% DMSO was used as vehicle control. Acrosome reaction was induced by incubating mouse sperm with 50 mouse solubilized zonae pellucidae for 15 min at 37 °C. Human sperm were incubated with 10 µM progesterone for 30 min at 37 °C. The sperm suspensions were sedimented by centrifugation at $2000 \times g$ for 5 min and the sedimented sperm were resuspended in 100 µl PBS buffer. Samples were air-dried on microscope slides and fixed for 30 minutes in 100% ethanol at room temperature (RT). For acrosome staining, mouse and human sperm were incubated for 30 min in the dark with 5 µg/ml PNA-FITC or 5 µg/ml PSA-FITC, respectively, and counterstained with 2 µg/ml DAPI. After curing, slides were analyzed using a Zeiss LSM 880 Laser Scanning Confocal Microscope; images were captured with two photomultiplier and one Gallium Arsenide Phosphide detector using ZEN Imaging software. For each condition, at least 600 cells were analyzed using ImageJ 1.52.

### Western blot analysis

Mouse sperm from vehicle- or inhibitor-injected mice were isolated 1 hour post injection from male mice injected (i.p.) with 150 µl of solution containing sAC inhibitor or vehicle. 'Swim out' sperm were diluted 1:1 to 1:100 in capacitating TYH media. As control, sperm from vehicle-injected mice were diluted 1:100 in non-capacitating and capacitating TYH buffer. The samples were incubated for 90 min, washed with 1 ml PBS and sedimented by centrifugation at $2000 \times g$ for 3 min. The sedimented sperm were resuspended in 15 µl 2× Laemmli sample buffer, heated for 5 min at 95 °C, supplemented with 8 µl β-mercaptoethanol and heated again for 5 min at 95 °C. For Western blot analysis, proteins were transferred onto PVDF membranes (Thermo Scientific), probed with anti-phosphotyrosine (1:1000, clone 4G10, Merck Millipore) and anti-mouse IgG (1:10,000, Cell Signaling Technology) antibodies, and analyzed using a chemiluminescence detection system. Image lab (Bio-Rad) was used for densitometric analysis of western blots.

### Sperm motility assays

For single-sperm motility analysis, mouse and human sperm tethered to a glass surface were observed in shallow perfusion chambers with 200 µm depth. An inverted dark-field video microscope (IX73; Olympus) with a 10 x objective (mouse sperm) or a 20 x objective (human sperm) (UPLSAPO, NA 0.8; Olympus) was combined with a high-speed camera (ORCA Fusion; Hamamatsu). Dark-field videos were recorded with a frame rate of 200 Hz. The temperature of the heated stage was set to 37 °C (stage top incubator WSKMX; TOKAI HIT). The images were preprocessed with the ImageJ plugin SpermQ Preparator (Gaussian blur with sigma 0.5 px; subtract background method with radius 5 px) and analyzed using the ImageJ plugin SpermQ[57]. The beat frequency was determined from the highest peak in the frequency spectrum of the curvature time course, obtained by Fast Fourier Transform.

For ex vivo assessment of mouse sperm from inhibitor-treated mice, sperm (25 μl) isolated at the indicated time points (15 min to 24 hours post-injection) diluted 1:5 in non-capacitating TYH buffer were loaded on a 100 μM Leja slide (Hamilton Thorne) and placed on a microscope stage at 37 °C. Sperm movements of 10 fields of at least 200 sperm were examined using computer-assisted sperm analysis (CASA) via Hamilton–Thorne digital image analyzer (IVOS II, Hamilton Thorne Research, Beverly, MA) with the following parameters: 30 frames, frame rate: 60 Hz, cell size: 30–170 μm$^2$. Ejaculated uterine mouse sperm were recorded without additional dilution. Sperm were considered motile when presenting STR ≥80% and VAP ≥50 μm/s and hyperactivated when presenting VCL ≥270 μm/s, LIN < 50%, and ALH ≥7 μm. Track´s color code: motile (green), progressive (turquoise), hyperactivated (purple), slow (pink), static (red), not counted due to late entry (yellow).

### Pharmacokinetic analysis
Male mice were administered 50 mg/kg TDI-11861 via oral gavage or injection as described above. At defined time points (15 min to 24 hours post-injection), 10 μl tail blood was progressively collected from the same males, mixed with 20 μl 15 mM EDTA, frozen in liquid nitrogen, and stored at −80 °C until analysis. Liver, kidney, eye, testis, caput and cauda epididymis, and seminal vesicle were collected, weighed, and homogenized in a threefold volume of 1:1 EtOH/H$_2$O (v/v). To assess the effects of chronic treatment with sAC inhibitor, 7 day (Alzet pump model 2001) and 42 days (Alzet pump model 2006) Alzet osmotic pumps were filled with 200 μl 50 mg/kg or 150 mg/kg TDI-11155, respectively. Mice anesthetized with ketamine/xylazine were shaved and residual hair was removed with hair removal cream. A small incision was made with a scalpel, a hemostat was used to create a pump-size pocket under the skin, and the pump was inserted with the pump opening facing away from the incision. After the wound was closed, mice were treated with meloxicam for pain relief. For 7 or 42 days post-surgery, tail blood was collected at regular intervals. The amount of TDI-11861 or TDI-11155 in each blood or tissue sample was quantified by LC-MS (Frontage Laboratories).

### Histology
Blinded histopathological evaluation of indicated organs (testis, epididymis, kidneys, eyes, liver, spleen, and pancreas) was performed by a board-certified comparative pathologist at the WCM Laboratory of Comparative Pathology. Three wild-type C57Bl/6 mice implanted with TDI-11155 (50 mg/kg) containing 7 day minipumps were compared to three wild-type C57Bl/6 mice implanted with vehicle-alone containing 7 day minipumps. Organs were fixed in 10% neutral buffered formalin followed by embedding into paraffin. Sections were cut followed by staining with hematoxylin and eosin (H&E) for microscopic evaluation.

### Clinical observation
Mice were monitored for 24 hours following TDI-11861 administration for the following clinical signs of abnormal behavior: convulsions, somasthenia, abnormal staying and moving posture, breathing, body temperature, body weight loss, mental state, condition of urine, feces, skin, and coat, and death.

### Mouse mating
Single-housed naive (i.e., uninjected and virgin) male and female C57Bl/6 mice were acclimatized to a reverse light cycle (dark: 11 AM to 11 PM) for at least 2 weeks. At 10:00 AM, males were injected (i.p.) with 150 μl sAC inhibitor solution or 150 μl vehicle control (DMSO:PEG 400:PBS 1:4:5 (v/v)). Thirty minutes or 1 hour later, individual injected males were paired with a female in estrus (identified by physical examination within the previous 30 min), and the pair were allowed to mate for the subsequent 2, 3, or 8 hours. Pregnancy and litter size was assessed in two ways. Either female were sacrificed 7 days following

mating and implanted embryos counted, or females were permitted to go to term (21 days) and pups counted. A subset of the pups (both male and female) born from breakthrough pregnancies were permitted to mature and their fertility was assessed in standard matings. To test fertility recovery after TDI-11861 injection, 24 hours after injection with 50 mg/kg TDI-11861, individual males were mated for 4 days with a female, and pregnancy (and litter size) was assessed after 21 days. In both vehicle-injected and TDI-11861-injected pairings, ~24% of females exhibited bona fide plugs.

### Statistical analysis
Statistical analyses were performed using GraphPad Prism 9 (GraphPad Software). All data are shown as the mean ± SEM, $n$ indicates the number of biological replicates. Statistical significance between two groups was determined using two-tailed, unpaired $t$ tests with Welch's correction, and statistical significance between multiple groups was determined using one-way ANOVA with Dunnett's correction. Differences were considered to be significant if $^*P < 0.05$, $^{**}P < 0.01$, $^{***}P < 0.001$, and $^{****}P < 0.0001$.

### Reporting summary
Further information on research design is available in the Nature Portfolio Reporting Summary linked to this article.

## Data availability
The primary data included in this study are provided in the Source Data File. Requests for materials should be addressed to Drs. Jochen Buck (email: jobuck@med.cornell.edu) and Lonny R. Levin (email: llevin@med.cornell.edu). Source data are provided with this paper.

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

## Acknowledgements

We wish to thank Sylvia Ayoub, Samuel Nidorf, and J.B.J. for assistance with mating experiments, Lee Cohen-Gould, Sushmita Mukherjee, and Janet Sun from the WCM microscopy core for assistance with microscopy, Harvey Florman for insights about long residence times, and Greg Kopf, Pablo Visconti, and the Levin/Buck laboratory for helpful advice on the manuscript. This work was supported by grants from the National Institutes of Health (P50 HD100549 to L.R.L., J.B., & P.T.M.; F31 AG069501 to T.R.; and F31 HD105363 to J.F.; and P30 CA008748 to I.C.M.) and Male Contraceptive Initiative (MCI) grants to L.R.L. & J.B., M.B., and C.R. The authors gratefully acknowledge the support provided by the Tri-Institutional Therapeutics Discovery Institute (TDI), a 501(c)(3) organization. T.D.I. receives financial support from Takeda Pharmaceutical Company, TDI's parent institutes (Memorial Sloan Kettering Cancer Center, The Rockefeller University, and Weill Cornell Medicine), and from a generous contribution from Mr. Lewis Sanders and other philanthropic sources.

## Author contributions

M.B., P.T.M., L.R.L., and J.B. conceptualized the study: M.B., T.R., J.F., R.W.M., C.S., and D.J.H. were responsible for the methodology: I.C.M. was responsible for the histology: M.B., TR, JF, and LG were responsible for the investigations: M.B. was responsible for visualization: L.R.L. and J.B. were responsible for supervision and funding acquisition: and M.B., C.R., L.R.L., J.B., P.T.M. were responsible for writing and editing the manuscript.

## Competing interests

Drs. Buck and Levin have licensed the commercialization of a panel of monoclonal antibodies directed against sAC to Millipore. Drs. Meinke, Steegborn, Levin, and Buck are co-founders of Sacyl Pharmaceuticals, Inc. established to develop sAC inhibitors into on-demand contraceptives. M.B., C.S., J.B., L.R.L., and employees of the Tri-Institutional Therapeutics Discovery Institute (TDI) may benefit from the further development and licensure of molecules described herein. All other authors declare no competing interests.
