## [Peer Review File · Nature Communications]

On-demand male contraception via acute inhibition of soluble adenylyl cyclaseEditorial Note: This manuscript has been previously reviewed at another journal that is not operating a transparent peer review scheme. This document only contains reviewer comments and rebuttal letters for versions considered at *Nature Communications* .

REVIEWER COMMENTS

Reviewer #3 (Remarks to the Author):

I have reviewed all of the responses to all four reviewers including my own questions. While I still believe that the manuscript would have been improved by better describing the features of the new molecular that led to such improvements I agree that it is probably beyond the scope of this manuscript. I personally think that the description of the structure in the previous paper that the authors cite is completely inadequate. If the coordinates are available then I guess one can make their own figure. For me it would have enhanced that paper. I am still concerned about the overlapping information in the other two manuscripts although I appreciate that it is important to get this data reviewed by the appropriate experts. Often when it is simply submitted as supplementary material it does not get properly reviewed if reviewed at all.

In summary, I will defer to the other three reviewers. I still think that this is an important "Proof-of-Principle" for drug discovery but I will go along with the other reviewers who raised more critical questions about the experimental design and executions. If they are satisfied, then I would concur with acceptance.

Reviewer #4 (Remarks to the Author):

The authors provided their explanations to the reviewer's comments and changes made to the manuscript, mainly by highlighting this study as the proof-of-concept study and by clarifying that none of the tested compounds are actually viable candidates as clinical compounds. As one of the related papers is now published and the other accepted, the publicly available information for the specificity of the two compounds are better referenced, which also slightly alleviates the concern about the readability of the current manuscript. Then, at the same time, certainly these changes to support their current conclusion and data decreases the impact of the study and novelty claim as the compound is still not a specific compound that can be used for male contraception in human just like those previously reported compounds, other than its demonstration of 'on-demand' concept, which is significant. While it is not this reviewer's suggestion, this reviewer finds adding phylogenetic tree of sAC and description of evolution of sAC vs. tmAC and sAC in sperm vs somatic cells suggested by reviewer 3 is very constructive and will dramatically increase the accessibility and readability of this manuscript for the broad audience of this journal while the work does not require any additional, wet experiment.

One major concern still remains regarding cAMP levels in ejaculated sperm (Major comment 8)

- Figure 4A and corresponding legends show the cAMP levels were measured from ejaculated sperm which were incubated in non-capacitating condition or capacitating condition further after harvested from the female tract. Considering successfully blocking fertility by IP injection to inhibit sAC function, this reviewer thinks even the ejaculated sperm incubated further at the BSA/HCO₃-negative condition (blue striped bar) should give lower sperm cAMP level. Moreover, the sperm from inhibitor injected males show compared level of cAMP compared to those from vehicle-injected males. Does this mean the IP-injected inhibitor does not successfully inhibit sAC blocking in female tract despite their acute exposure to the bicarbonate and albumin as the authors mentioned in their response? If that is true, then the observed infertility phenotype from nature mating cannot be directly blocking sAC?

Response to reviewer critiques:

Reviewer #3

I still believe that the manuscript would have been improved by better describing the features of the new molecular that led to such improvements.

The revised manuscript now includes Figure 1B, which shows an overlay of the sAC crystal structures in complex with the two inhibitors discussed (TDI-10229 and TDI-11861), and a short description of the molecular features rationalizing the higher potency of TDI-11861. To accommodate these changes, we have added two authors (DJH and CS).

Reviewer #4

this reviewer finds adding phylogenetic tree of sAC and description of evolution of sAC vs. tmAC and sAC in sperm vs somatic cells suggested by reviewer 3 is very constructive and will dramatically increase the accessibility and readability of this manuscript for the broad audience of this journal while the work does not require any additional, wet experiment.

We revised the manuscript to include a description of the evolutionary relationship between sAC and tmACs, and because we recently published a perspective detailing the relationship between sAC in sperm versus sAC in somatic cells, we summarized those conclusions and included a reference to our recently published paper.

One major concern still remains regarding cAMP levels in ejaculated sperm (Major comment 8) - Figure 4A and corresponding legends show the cAMP levels were measured from ejaculated sperm which were incubated in non-capacitating condition or capacitating condition further after harvested from the female tract. Considering successfully blocking fertility by IP injection to inhibit sAC function, this reviewer thinks even the ejaculated sperm incubated further at the BSA/HCO₃⁻ negative condition (blue stripped bar) should give lower sperm cAMP level. Moreover, the sperm from inhibitor injected males show compared level of cAMP compared to those from vehicle-injected males. Does this mean the IP-injected inhibitor does not successfully inhibit sAC blocking in female tract despite their acute exposure to the bicarbonate and albumin as the authors mentioned in their response? If that is true, then the observed infertility phenotype from nature mating cannot be directly blocking sAC?

We apologize for not being clearer; we did not fully appreciate the reviewers concern in making our previous response. When we collect ejaculated sperm from the female reproductive tract, we also collect a considerable number of other cells and debris which originate from the female reproductive tract. If you look carefully at the images in Figure 4B, you can see that there are a other cells which are not sperm. Presumably they are immune cells, but most importantly, they are not derived from sAC inhibitor treated males. Thus, the basal cAMP levels measured in non-capacitating conditions in the samples collected from the female reproductive tract reflects a mix of cAMP levels in female (uterine) immune cells which have never seen sAC inhibitor and ejaculated sperm. In contrast, incubating this mixture of female immune cells and ejaculated sperm in bicarbonate/BSA selectively stimulates cAMP production in the ejaculated sperm cells, and as the data in Figure 4A show, there is no stimulation due to capacitating conditions in the sperm from sAC inhibitor treated mice. We have revised the methods section to alert the reader to this issue.